# Improving Participation among Youth with Disabilities within Their Unique Socio-Cultural Context during COVID-19 Pandemic: Initial Evaluation

**DOI:** 10.3390/ijerph20053913

**Published:** 2023-02-22

**Authors:** Anat Golos, Chani Zyger, Yael Lavie-Pitaro, Dana Anaby

**Affiliations:** 1School of Occupational Therapy, Faculty of Medicine, Hebrew University, Jerusalem 91240, Israel; 2School of Physical and Occupational Therapy, McGill University, Montreal, QC H3G 1Y5, Canada

**Keywords:** participation, quality of life, disabilities, socio-cultural context, environmental factors, COVID-19 pandemic

## Abstract

Background: Participation in meaningful activities is important for quality of life among youth with disabilities; yet participation is often restricted during adverse times. This study aimed to explore the effectiveness of the Pathways and Resources for Engagement and Participation (PREP) intervention among ultra-Orthodox Jewish Israeli youth with disabilities during the COVID-19 pandemic. Methods: A 20-week single-subject research design with multiple baselines measuring participation goals/activities was employed with two youths (15 and 19 years old) combining quantitative and qualitative descriptive data. Changes in participation levels were measured biweekly using the Canadian Occupational Performance Measure; COPM; participation patterns were measured using the Participation and Environment Measure—Children and Youth; PEM-CY pre- and post-intervention; parents’ satisfaction was measured using the Client Satisfaction Questionnaire, CSQ-8. Semi-structured interviews were conducted post-intervention. Results: Both participants significantly improved participation in all selected goals and participation patterns, and were highly satisfied with the intervention. The interviews revealed additional information on personal and environmental barriers, factors supporting intervention, and intervention effects. Conclusions: The results indicated that an environment-centered and family-centered approach can potentially improve the participation of youths with disabilities within their unique socio-cultural context, during adverse times. Flexibility, creativity, and collaboration with others also contributed to intervention success.

## 1. Introduction

Participation in meaningful activities is recognized as a significant factor affecting human health, wellbeing, and overall quality of life [1]. Participation, according to the International Classification of Functioning, Disability, and Health (ICF) [2], is defined as involvement in a life situation; it is considered an important outcome measure for rehabilitation and intervention [1]. It is a multidimensional concept that includes the objective aspect of frequency, and the subjective aspect of involvement [3], which can be described as the personal experience of participation.

The participation of children and youth with disabilities was found to be limited, compared to their typically developing peers [4]. They participate less frequently and/or are less involved in activities in different settings, compared to their peers without disabilities [5,6]. This limitation was also noted in social and leisure activities, such as organized physical activities in the community [7]. Participating in leisure activities is important for youth with disabilities, since such participation can potentially develop their domains of competence and interest, and enhance their social interactions with peers and with the wider community [8]. Participation in community-based leisure activities is an important outcome of occupational therapy intervention, which was found to be significantly restricted among youths with various types of disabilities in North America [7], as well as in Israel [9].

Participation in meaningful activities occurs in different environments, such as school, home, and community [2]. Environmental factors may also affect a person’s development and participation [10,11]; they can either support or hinder participation of children and youth with disabilities [12]. In particular, positive attitudes, social support, and availability of adapted services can serve as environmental supports for youth with disabilities [10]. Environmental support, however, was found to be significantly lower among children and youth with disabilities such as those with attention deficit hyperactivity disorder (ADHD), which especially impacted their involvement in different settings [13].

One of the interventions that focuses on environmental factors and their impact on participation is the Pathways and Resources for Engagement and Participation (PREP), an approach developed by a group of Canadian occupational therapist (OT) researchers [12]. This is a client-centered, evidence-based approach to occupational therapy that seeks to enhance participation by modifying the environment. The PREP is designed for use in the places where the participants live, learn, and play: the home, school, and community. Using this approach, OTs work with clients and their parents or caregivers to identify aspects of the environment (physical, social, attitudinal, and institutional) that either support or hinder participation in a desired activity. Together, a plan is developed and implemented to minimize and/or remove barriers in the environment and build on existing supports, so that the clients can participate in the activities of their choice. It involves five steps: “make goals”, “map a plan”, “make it happen”, “measure process and outcomes”, and “move forward”; implemented in three phases: (a) a baseline targeting three chosen goals (weeks 1–4); (b) intervention for a period of 12 weeks (weeks 5–16), with a four-week intervention designated for each goal; and (c) follow-up (weeks 17–20).

Previous studies that examined PREP’s effectiveness indicated improvement in the participation patterns in selected activities among children and youth with disabilities, achieved by targeting and applying changes to various environments (i.e., home, school and community) and/or to the selected activities (e.g., joining a drawing class, playing a musical instrument, attending spiritual/religious gatherings). Participants’ parents also reported high levels of satisfaction with the improvement of their child’s participation, as well as with the overall service they received during the intervention [14,15,16,17]. Improvements in specific body-function-level outcomes (motor, such as strength; cognitive, such as attention; and emotional, such as self-esteem) were also found following the PREP intervention [18].

Moreover, parents reported significant improvement in their children’s participation and independence. One of the main factors they mentioned that contributed to the success of the intervention was the OTs, who guided and assisted them in finding activities that encouraged independence [19]. A recent study was conducted in Ireland [17] among three boys (6–7 years old) with functional difficulties, who had been born prematurely. They reported that the children’s performance of selected activities significantly improved following 12 weeks of PREP intervention. These findings demonstrate the potential impact of the PREP approach in improving participation among a population with diverse ages, disabilities, and living environments.

While previous studies support the effectiveness of the PREP, they were conducted mostly among English and French speakers in different regions worldwide (e.g., Canada, UK, Australia), with no reference to cultural or religious community influences. Therefore, it is worthwhile to examine its effectiveness among populations with disabilities from different socio-cultural backgrounds. One population with a unique socio-cultural background is the Jewish ultra-Orthodox (UO) community, a fast-growing population in Israel and the world. They account for approximately 10% of the overall population in Israel and are estimated to triple in number over the next three decades [20].

The UO community is a highly religious group with a commitment to *halacha*, a body of Jewish laws and customs [21] which serves as a central guide for the community. Their values, beliefs, and occupations are greatly influenced by their community’s cultural codes and obligations, which influence all aspects of daily living [22]. They highly value religious learning, and the social criteria by which they rate adolescents at various life junctions typically includes acceptance to a UO educational institution [23]. UO members usually prefer to live in segregated communities, with separate educational institutions and leisure community centers. Their cultural conservatism is reinforced with fixed boundaries between themselves and the general population, in order to minimize outside influences. Thus, UO members have less accessibility to the Internet and other resources [24]. Additionally, they tend to have larger than average families, and lower than average incomes [25,26]. Therefore, UO cultural attitudes and socioeconomic status commonly affect their participation in daily activities, as well as their quality of life, health, and community services. For example, a recent study [27] that explored the live experiences of mothering a child with ADHD, using semi-structured in-depth interviews, found that the perception of their maternal role, social factors, and legitimacy of self-care highlighted the unique impact of the UO culture on the mothers’ lived experience. The findings of the current research relating to the UO community [24,27] can likewise highlight the need to promote culturally sensitive health care and interventions for the UO population.

During the COVID-19 pandemic, implementation of the PREP protocol was adjusted to enable participation in desired activities among youth with disabilities under the prevailing restrictions. The benefits of participation generally become more evident during extraordinary times; yet the COVID-19 pandemic restrictions required staying at home more and following social distancing rules. It was difficult for children and youth to understand and follow these rules, especially those with disabilities [28]. This changing pandemic situation required flexibility and creativity in implementing occupational-therapy interventions, such as the PREP protocol, in order to make them accessible to youth with limited participation [29]. In addition to its impact on participation, the pandemic also negatively impacted the quality of life and mental health of children and youth [28,30]; and those with low socioeconomic status and limited living space were significantly more negatively affected [30]. This was also noted in tracking the wellbeing of children with neurodevelopmental disabilities [31]. Children and youth with disabilities, as well as people with chronic health conditions, and members of minorities were all considered vulnerable to the pandemic [32].

The unique cultural codes and values of the UO community and their lower socioeconomic status, both of which affect their participation in daily activities even in normal times, create special vulnerability among UO youths with disabilities. The adverse and changing conditions of the COVID-19 pandemic further exacerbated that negative impact. Therefore, the UO community was chosen for focused study, in order to expand the examination of the PREP intervention effectiveness among different socio-cultural populations under diverse conditions. This study, therefore, aimed to provide initial evidence of the effectiveness of the PREP approach for UO youths with physical disabilities during the pandemic.

## 2. Materials and Methods

### 2.1. Study Design

A combination of quantitative descriptive and qualitative data was used, to produce richer and more comprehensive research results based on the intervention reports of the participants and their parents, as well as on the experience and perspectives of the participants and their participation team (family and community members involved in the intervention). In this study, qualitative descriptive data were used to complement the quantitative findings, producing deeper insight into the intervention process and outcomes. The quantitative information was collected using a single-subject design with multiple baselines across participation goals/activities, involving a repeated-outcome design. The effect of the intervention was examined across three leisure activities/participation goals for each youth; each participant served as their own control [33]. The qualitative information was collected following the completion of the intervention, through semi-structured in-depth interviews conducted individually with each of the participants and their participation team (family and community members involved in the intervention).

### 2.2. Participants

Participants were recruited using a convenience sample through UO community representatives, who approached UO families of youths with disabilities to solicit their participation in the study. Families who showed interest contacted the researcher; following an oral explanation and on meeting the inclusion criteria, participants received an information sheet and signed an informed consent form. Four families showed interest, while three youths were found to be suitable. One family later withdrew due to the COVID pandemic.

Ultimately, participants included two youths, a 15-year-old male and a 19-year-old female. The inclusion criteria for participation in this study were youth with physical disabilities and verbal communication abilities, and consent from them and their parents to participate in this study. The exclusion criteria were a medical condition that does not allow them to leave home, and attendance in a special educational setting. Both participants were diagnosed by a medical professional with spina bifida, a developmental malformation of the spinal cord that leads to complications in several organ systems and considerable disability [34]. They both used a power/manual wheelchair for mobility and attended a mainstream inclusive high school educational setting within their community. They reported having motor, emotional, and social difficulties, as well as taking medication regularly. According to the Functional Independent Measure (FIM) (see the Instruments section), the first participant needed minimal assistance (score 78/126), and the second participant was independent (score 97/126). Both participants lived in a UO city in ground-floor apartments with adequate access to their homes. However, they needed assistance with moving around outside their homes, due to the lack of accessibility in their neighborhood.

### 2.3. Instruments

#### 2.3.1. Demographic Questionnaire

This questionnaire was developed for this study, in order to provide demographic and medical information about the participants, such as age, sex, and health condition.

#### 2.3.2. Canadian Occupational Performance Measure (COPM)

The COPM is a standardized assessment that uses a semi-structured interview to guide parents and youth in identifying activities and goals that are important to them but are difficult to carry out, and to rate their level of performance on a 10-point scale (1 = “unable to perform” to 10 = “performs extremely well”) [35]. Using the COPM twice a week, from the baseline, throughout the intervention, and after intervention, enables researchers to measure the change following the intervention. An increase of at least two points indicates a clinically significant change in performance, serving as an excellent responsive outcome measure [35,36]. The COPM has demonstrated reliability and validity, such as test–retest (r = 0.88–0.89), as well as the ability to detect changes in performance over time. In this study, it was used to determine intervention goals, as well as progress during the intervention and at the concluding follow-up.

#### 2.3.3. The Participation and Environment Measure—Children and Youth (PEM-CY)

The PEM-CY is a parent-report instrument that examines participation patterns and environmental factors affecting the participation of children and youth (5–17 years of age) [37]. It assesses participation in 25 activities across three settings: home (10 activities), school (5 activities), and community (10 activities). Parents rated their child’s participation frequency (1 = “never” to 7 = “daily”), level of involvement (1 = “minimally involved” to 5 = “very involved”), desire to change the child’s participation, and environmental barriers/supports affecting participation. The PEM-CY is a reliable and valid measure, with moderate to very good internal consistency (α = 0.59 to 0.91) and test–retest reliability: ICC = 0.58 to 0.95 [38]. In this study, the home and community sections (frequency, involvement, and desire for change) of the PEM-CY were completed at baseline and at the concluding follow-up. The Hebrew version, effectively used among the Israeli population [13], was administered.

#### 2.3.4. Functional Independence Measure (FIM)

The FIM is a functional status instrument for use among rehabilitation in-patients over the age of 7 [39]. It is used for assessing the performance of basic daily activities (BADL) using a 7-point scale (1 = “the patient needs maximum help;” 7 = “completely independent”). It includes motor and cognitive sections. The overall score is between 18 and 126, where higher scores indicate higher degrees of independence. The FIM has a high internal consistency (α = 0.93), high reliability between raters (r = 0.94), adequate discriminative capabilities for rehabilitation patients, and responsiveness over intervention [40]. In this study, it was administered at baseline to assess the functioning levels of the participants.

#### 2.3.5. Client Satisfaction Questionnaire (CSQ-8)

The CSQ-8 is used to measure levels of satisfaction with a program or service, using eight items rated on a 4-point scale (1 = “minimal satisfaction” to 4 = “maximal satisfaction”) [41]. A summary mean score ranges from 8 to 32, where higher scores mean higher satisfaction. It is a valid and reliable tool with high internal consistency (coefficient α = 0.93) [42], and construct validity; it has been used among families of children with physical disabilities [41]. In this study, the CSQ-8 was completed post-intervention by the parents.

#### 2.3.6. Semi-Structured Interviews

Individual semi-structured interviews were conducted for an hour at the end of the intervention in order to gather qualitative information from both participants and their participation team (a total of 7 interviews). The participation team included family and community members involved in the intervention. For Participant 1, the team included his mother, his adult married sister, who lived nearby, and an adult UO tutor recommended by the city welfare department. For Participant 2, the participation team included her mother and an UO woman who teaches art classes in the same neighborhood. Interviewees were asked to describe their overall experience and perspectives relating to the intervention process, based on an interview guide developed by two of the study authors following the PREP protocol (e.g., “Describe your general experience with the intervention process.” “What were the supporting and hindering factors for the intervention?” “Does the intervention seem to have affected the participant? If so, please describe.” “What contributed to the achievement of the goals?” “Was the occupational therapy involvement beneficial to you? If so, please describe.”).

### 2.4. Procedure

This study was approved by the Ethics Committee of the Hebrew University (No. 08082019). Following the recruitment process (as described in the Participation section) two UO youths, who along with their families consented to participate in this study, were included. The occupational therapist (OT) who implemented the intervention met each participant, mostly in person, according to the PREP protocol [12], once a week for a period of 20 weeks; meetings were divided into three phases: baseline (weeks 1–4), intervention (weeks 5–16), and follow-up (weeks 17–20).

#### Content and Intervention Process

At the beginning of the baseline phase, the OT met with each participant and family in their homes to complete a demographic questionnaire and the PEM-CY, and to identify occupational areas of priority, using the COPM to focus on what was meaningful and important to them. Then, the OT asked them to select three leisure activity goals at home and/or in the community, in which the youth wanted to participate but found it difficult. The specific goals that were identified were related to the participant’s socio-cultural context and roles, such as preparing and cooking meals for family members and participating in the community’s religious activities (see Table 1). During this phase, the participants together with their parents rated their performance concerning each goal twice a week, using the COPM performance scale.

Following this phase, the PREP intervention was introduced. For each goal, the OT, parents, and participants jointly identified barriers in the physical environment (such as outdoor areas not accessible for wheelchairs) and in the social environment (such as others who do not encourage them to experience new activities). Barriers were also identified in the activity’s demands (such as the need to know how to choose and purchase products, and how to manage money for payment), as well as the participant’s emotional barriers (such as fear of new experiences, or fear of other people’s reactions). They also explored solution-based strategies to modify these barriers, while building on existing supports in order to facilitate participation; for example, identifying and harnessing community resources (accessible shopping centers and synagogues in their area of residence). Additionally, the OT recruited a “participation team” for each participant, consisting of people involved in the intervention (e.g., family members, a tutor, and an art teacher).

The intervention period was 12 weeks, with four weeks or more devoted to each goal. A flexible implementation of the intervention protocol was necessary because of changing needs in the unpredictable circumstances of the COVID-19 pandemic, including imposed closures that kept the participants at home and prevented face-to-face meetings. As a result, a few sessions were conducted by phone (Internet communication is not commonly used among the UO population, due to religious scruples). In addition, one of the participants was hospitalized for a few days, due to a deterioration in their health condition, so a short break was required.

During the intervention, the OT encouraged and supported the participants, seeking to enable participation in chosen activities, devising strategies, and analyzing their accessibility needs in order to participate (e.g., rating and construction of meal plans, and discussing conversation topics prior to social interaction). The OT shared information and strategies with the participation team, and guided them in how to accompany, support, and encourage the participants in practicing the activities’ goals during the intervention. Participants’ performance regarding each goal continued to be monitored biweekly by the OT, while participants together with their parents rated their performance concerning each goal twice a week, using the COPM performance scale throughout the intervention.

Treatment fidelity was ensured by having the OT complete a workshop on the PREP approach, as well as ongoing expert consultation throughout the study. For each activity, intervention procedures and strategies were documented by the OT, using a structured form to ensure that the therapy was guided by PREP principles. Additional circumstances that may have had an effect on the intervention were also documented.

During the follow-up, which lasted 4 weeks after completion of the intervention for all three goals, participants and their parents continued to rate the participant’s performance levels for each goal twice a week, using the COPM. At the end of this phase (at week 21), the OT asked each participant and family to complete the PEM-CY and the CSQ-8. In addition, semi-structured interviews were conducted by another UO researcher who was not involved in the intervention process, for an hour individually with each of the participants and their participation team (overall 7 interviews); these were recorded and then transcribed.

### 2.5. Data Analysis

#### 2.5.1. Quantitative Analysis

The SPSS version 26.0 (Statistical Package for the Social Sciences) [43] was used for quantitative analysis. Descriptive statistics were used to describe the participants’ levels of independence at baseline, using the FIM, and the PEM-CY scores pre- and post-intervention.

For each participation goal, a series of data points representing the level of goal performance, generated by the COPM, was plotted and analyzed to detect change. Visual inspection was conducted to identify an increase of at least two points on the COPM scale, indicating a clinically significant change [44]. To detect changes that were statistically significant, the celeration line (CL) technique was used, obtaining a baseline for each goal, which represents the participants’ pre-intervention scores. This method calculates the proportion of data points falling above and below the CL; when more than 50% of data points are above the line, a change in goal performance is identified, indicating an intervention effect [45].

#### 2.5.2. Qualitative Analysis

Qualitative descriptive content analysis was employed to analyze the data from the participants’ interviews. This method is a process of describing qualitative data that results in clusters of responses. It involves establishing categories and sub-categories that refer to the manifested content [46]. The interviews were recorded and transcribed, and the text was analyzed using the principles of content analysis [47], which employ ongoing discussions regarding the data and comparative analyses. Two authors separately read the transcriptions to become familiarized with the data, assigned initial coding to meaningful units in the interviews, and then compared codes. Discrepancies led to a discussion, and the final codes were collected into categories and sub-categories that emerged from the semi-structured interview questions relating to the PREP intervention. The authors conducted discussions regarding the categories and sub-categories until agreement was established, using a coding tree to visualize them. Finally, relevant quotes from the dataset were selected that represent the interviewee’s perceptions regarding the intervention process.

## 3. Results

### 3.1. Changes in Performance Scores (COPM)

Quantitative analysis of the changes in performance scores was conducted before, during, and following the intervention, using a descriptive visual analysis of the COPM performance scores for all six goals (three goals for two youth). As shown in Figure 1, all goals were initially rated 1 (“unable to perform”) on the COPM scale, and this rating did not change during the entire baseline phase (weeks 1–4). However, during the intervention (weeks 5–16), there was a gradual increase in the performance scores for each goal; the scores at the post-intervention phase showed an increase of 6–8 points on the COPM scale. These results surpassed the clinically significant threshold of 2 points for improvement in performance, and they were consistently seen in all goals during the intervention phase.

In addition, all data points during and after intervention (100%) fell above the CL, the stable baselines observed pre-intervention. The higher scores were also maintained during the follow-up, with an additional improvement reported at week 21 in the second goal of the first participant. These results indicated significant changes in the performance scores in all selected goals, relative to the CL, which may support the effectiveness of the PREP intervention. 

### 3.2. Level of Satisfaction

Parents reported improved satisfaction with their children’s performance at the post-intervention phase (an average improvement of 7.67 points on the COPM satisfaction scale). Additionally, they reported maximum scores of 32 points on the CSQ-8, which indicated very high satisfaction with the intervention.

### 3.3. Changes in Participation Patterns

The results of the pre- and post-intervention PEM-CY scores indicated positive changes in participation patterns as reported by parents. Improvement was seen in the average scores of both participants in frequency (Home: from 3.9 to 5.5 out of 7; Community: from 1.6 to 2.7 out of 7), and in the level of involvement (Home: from 3.1 to 4.6 out of 5; Community: from 2.0 to 4.4 out of 5). Additionally, there was a decrease in the average percentage of activities in which parents desired change (Home: from 57.2% to 52.2%; Community: from 90% to 65%), as well as a decrease in the average number of barriers to participation reported by parents after intervention (Home: from 2.5 to 0, Community: from 6 to 4.5).

### 3.4. Qualitative Analysis

Using the qualitative information gathered through interviews with each participant and his/her participation team (Participant 1, his mother, sister and tutor; Participant 2, her mother, and an art teacher), four main categories emerged: (a) personal barriers; (b) environmental barriers; (c) factors supporting intervention; and (d) intervention effects (see Table 2). These categories provided additional information on the participant’s experience and perspectives relating to the intervention process, with additional explanations for the changes observed using the quantitative measures.

#### 3.4.1. Personal Barriers

The personal barriers related to the obstacles the participants faced at the beginning and throughout the intervention, in three sub-categories.

(A) Social barriers manifested as difficulties in communicating with peers, as described by Participant 2: “*I talk less with friends. I did not talk on the phone at all—only in class. After I chose a girl to call, I needed to think about what to talk about with her*”.

(B) Emotional barriers to participation were characterized as low self-efficacy and low motivation, as described by Participant 1’s sister: “*He had low motivation and low self-confidence. He did not believe in himself that he was capable of doing anything*”.

(C) Cognitive barriers were factors affecting their understanding of an activity’s cognitive demands, as described by Participant 1’s tutor: “*He did not know how to drive safely with the electric wheelchair on the sidewalk, and he did not know what to do in order to cross the road safely*”.

#### 3.4.2. Environmental Barriers

The environmental barriers included three sub-categories that limited participation in different settings.

(A) Physical barriers in the environment were those that hindered wheelchair accessibility, as described by Participant 1’s mother: “*There are not enough accessible sidewalks, and then he does not know what to do*”.

(B) Social barriers related to the human environment that surrounded the participants, including attitudes of family and community members, as described by Participant 1’s sister: “*They [the parents] spoiled and protected him too much, and he stayed that way; mostly Mom, but everyone was affected by it*”. Participant 1’s tutor commented: “*His mother also needs to change her attitude towards him, trust him more*”.

(C) Cultural barriers were related to the UO community, as described by Participant 2’s mother: “*We had to convince her to go to the mall, which is not acceptable in our community due to our religious values. She did not want to go, but then we concluded that if she wants to choose clothes for herself, it is possible only in the mall*”.

#### 3.4.3. Factors Supporting Intervention

The factors that supported the intervention included five sub-categories.

(A) Encouragement and support were provided by the OT who implemented the intervention, and others, as described by Participant 1: “*The OT gave me the courage, and the tutor continued it*”. Participant 2’s mother said: “*The encouragement given by the OT helped us*”.

(B) Mediating and imparting *knowledge and strategies* were provided by the OT, as described by Participant 1’s mother: “*The mediation that the OT gave him helped. At first, she went with him and showed him all the accessible ways, and he gained confidence*”. Participant 2′s mother observed: “*The OT taught her to write a recipe in an accessible way for her*”.

(C) Adaptation of the environment and/or activity for accessibility was described by Participant 2’s mother: “*The kitchen has changed and is much more accessible to her. She has learned to edit recipes in her own way*”.

(D) Practicing activities with others was described by Participant 1’s tutor: “*We practiced the ride and crossing the road, and he has made great progress*”.

(E) Acceptance by others was described by Participant1’s mother: “*The rabbi in the synagogue accepted him. That is the reason he wants to go there; they do not think he is so different*”.

#### 3.4.4. Intervention Effects

The effects of the PREP intervention on the participants and their families included five sub-categories.

(A) Improved independence was noted as described by Participant 1’s tutor: “*He [P1] started going out to the synagogue alone, crossed the road alone, entered alone, and stayed there for prayer and for a class*”. His sister noted: “*Before the intervention, the family asked us to come and look after him and sleep with him. Today he sleeps alone if needed, and if Mother goes out, he also stays alone*”.

(B) Improved initiative in activities was seen in both participants. Participant 1’s mother: “*One day he decided he was going to the synagogue alone; it was something new*”. Participant 2’s teacher: “*She initiates a lot of social conversations with the other girls*”.

(C) Feeling happier and more enjoyment was described by Participant 2: “*When we organized the Shabbat meal with friends, I made a larger amount. When I do something for others, I feel like everyone else. It makes me happy*”. Participant 1′s tutor: “*When we finish a successful purchase at the supermarket, or when we leave the synagogue, he looks happier. There is a light in his eyes when I come. He shares with me, and you can see how happy he is*”.

(D) Improved confidence and sense of competence was described by Participant 1’s tutor: “*I see more confidence. At the beginning, he apologized for everything... and* today it is better”. Participant 2 remarked: “*The process helped me and gave me confidence to do things*”.

(E) Improved communication with others, such as family members and peers, was described by Participant 1’s sister: “*Today, he is a boy with desires; he initiates communication both with me and other family members, and expresses himself*”. Participant 2’s teacher: “*She initiates more phone calls to friends. She talks a lot with friends, not necessarily with her classmates*”.

## 4. Discussion

This study provides initial evidence supporting the effectiveness of the PREP approach among youths with disabilities in the UO community, specifically during the COVID-19 pandemic when additional circumstances affected participation in activities. The study involved an in-depth examination, using both quantitative and qualitative descriptive data, which provided complementary information. This enabled us to better understand the implementation and test the effectiveness of the PREP approach in the unique cultural context of the UO community, during challenging times. The following discussion integrates both qualitative and quantitative data.

The results indicated significant improvement in participation in all participants’ and their parents’ six selected goals, as measured by the COPM and tested using the CL technique, which supported the effectiveness of the PREP intervention. These findings coincide with previous studies examining the effectiveness of this intervention among youth with physical disabilities. For example, a significant improvement was found in the performance of most of the selected goals following intervention among six Canadian adolescents with physical disabilities (e.g., spina bifida and cerebral palsy) [14], as well as among 28 adolescents with moderate physical disabilities [15].

Regarding the improvement that we reported, a possible explanation emerged in the interviews that this could be related to the involvement of others in the intervention process, including family members. This explanation is supported by previous research conducted among occupational OTs, who reported that parental involvement was an important factor contributing to intervention success, since the parents promoted procedural aspects such as use of resources and transportation [48].

In our study, improvement was observed throughout the intervention at different time points and in parallel with a number of goals. As a result, the intervention focus was sometimes on more than one goal at a time, and each goal period lasted longer than planned. In addition, some sessions were conducted over the phone rather than face-to-face, due to the pandemic restrictions. The participants’ improved performance of their selected goals, despite these barriers, may suggest that the intervention is effective even during times of adversity, although flexibility and creativity are necessary. The improvements in participation measured by the COPM were maintained in the follow-up phase, which may further confirm the effectiveness of the intervention, as noted in a previous study [15].

The overall participation patterns in the home and the community, in terms of both frequency and involvement, were also improved following the intervention, as measured by the PEM-CY. These results can be explained by the fact that the intervention focused on participation rather than on specific functions, and because intervention was conducted in the participants’ natural environments. These findings concur with previous studies that found improvement in participation in different domains and settings among youth aged 12–18 with physical disabilities [14,49].

The qualitative findings gleaned from the interviews revealed factors that can explain the improvement in participation, as evidenced by the changes in COPM scores. The main factor was the occupational OT’s role and the strategies she used during the intervention, including encouragement, support, mediation, and sharing information (knowledge and strategies) with the family and participation team: how to engage in activities, while considering accessibility; how to adapt the environment and activities; and how to encourage acceptance by others. These findings support previous studies highlighting the OT’s role in guiding and directing the intervention process [19,48].

Our participants faced personal barriers to participation, including social barriers (difficulties in communicating with peers), emotional barriers (low self-efficacy and motivation), and cognitive barriers (difficulties in understanding the activity’s demands). These barriers were not treated directly, but rather by providing environmental supports (such as a tutor selecting or rating activities appropriate for the youth). This was also noted in a previous study among OTs, indicating that changes in the social barriers during the intervention led to greater participation [48].

The participants also faced environmental barriers to participation, including physical and socio-cultural barriers. The barrier presented by the socio-cultural environment was similar to the findings of a previous study, in which people with disabilities reported that unsupportive attitudes and a stigma attached to different cultures can be a barrier to participation in their workplace [50]. These findings highlight the impact of socio-cultural norms on the participation of people with disabilities in different environments. Our study was conducted among the UO community, where it was found, for example, that social stigma may contribute to the tendency of UO parents to conceal their child’s disability [24]. An additional environmental barrier was a technical limitation in maintaining contact during pandemic restrictions, due to the tendency of many UO families to avoid using high-tech communication channels such as Zoom or WhatsApp [24]. In our study, communication with participants and their families was needed throughout the intervention despite the barrier; therefore, phone calls became the preferred option.

Several UO cultural characteristics also served as factors supporting participation. The UO members usually live in segregated communities [25], with a central role given to the family and community. According to the interviews, the involvement of family and UO community members, who lived near the participants in the intervention process, contributed to the success of the intervention. This was also noted in the study of Golos et al. (2021) [24], which reported the need perceived by OTs for adapting an intervention protocol for UO children with ADHD, and suggested considering the lifestyle and cultural values of this community, as this contextual factor may affect the child’s routines and participation.

The physical environment of the public space in UO neighborhoods has also been reported as a barrier to the participation of individuals with disabilities, including a lack of accessible sidewalk ramps which limits the mobility of wheelchairs. Given that the UO population often lives in separate neighborhoods or cities, with large families and low incomes [25], they often face poor infrastructure, including a lack of policies to support accessibility. In addition, this community offers few cultural activities accessible to people with disabilities; in our study it was possible—but difficult—to find suitable settings for the participants. Moreover, most community settings were shut down during the COVID-19 pandemic; therefore, the intervention was necessarily built on the remaining available settings (home, the supermarket, and the synagogue).

Following intervention, the interviewees reported improvement in the participants’ self-confidence and competence, their initiation of activities, their communication with others, their sense of enjoyment, and their growing independence. These findings concur with a previous study that reported improvement in self-confidence, self-esteem, and even physical health among youth with disabilities following intervention [19]. In addition, the results of our study indicated a significant improvement in parental satisfaction with their children’s performance, and with the service they received. These results are not surprising, since the parents directly observed the progress of the participants, as well as changes in their behavior in life domains beyond the selected goals. Parents also appreciated the quality of personal relationship with the OT, as well as the encouragement, support, knowledge, and strategies that were provided by the OT. These findings are similar to those of previous studies that reported high satisfaction with the service provided to youth through this approach [14,44].

In this study, the COVID-19 pandemic affected the participants’ lives, which is plausible since the impact of pandemic restrictions was harder on children and youth with disabilities, compared with their typical peers [28]. The pandemic also affected the content and execution of the intervention; therefore, implementing the PREP protocol with the two participants required flexibility and creativity, as noted in the literature [29]. First, the selected goal needed to be adapted to the pandemic restrictions, which presented limited opportunities for socializing in community activities. In addition, since most community settings were shut down during this period, the intervention needed to build on an available setting, such as the home and small community settings. For example, one of the selected goals of one participant was to maintain social interaction in meeting with peers. This could be implemented only after recruiting an art teacher, who could conduct a joint creative activity for a small group of peers in her home. The second participant chose praying at the synagogue as one of his intervention goals, mainly because this setting was available due to its importance for the UO community. This choice may also confirm the compensatory influences of religion and spiritual activities in critical situations, which were found to improve mental health outcomes during the pandemic [51]. Second, due to the tendency of many UO families to avoid using high-tech communication channels, communication with the participants was sometimes possible only through phone calls. In addition, the number of sessions and duration of some goals were changed; thus, the intervention did not correspond exactly to the protocol schedule.

However, despite the effect of the pandemic on the overall intervention, the results indicated improvements in the performance of selected goals and participation patterns. Nevertheless, the impact of the pandemic on the outcomes needs to be further evaluated in comparison to results of post-pandemic intervention among this population.

### 4.1. Clinical Implications

Our findings lend further support for focusing on a range of environmental barriers in order to improve participation in occupational therapy practice, rather than focusing solely on changing impaired body functions (such as motor and cognitive abilities). This was likewise suggested in a systematic review by Novak et al. (2013) [52]. In times of adversity, flexibility and creativity are also needed during intervention to identify activities and find solutions together with other stakeholders in the community. This may include changing the number of sessions and duration of some goals, and/or conducting a few sessions remotely, even by phone. Although the intervention can be feasibly adapted for the needs of the target population, it should be modified with regard to the available resources, particularly in times of adversity. Finally, natural environmental interventions should be considered, which preserve the values and needs specific to the participants’ socio-cultural context, make use of the appropriate resources and accessible settings available in the community, and assemble a participation support team from community members.

### 4.2. Limitations and Recommendations for Further Research

The present study has a number of limitations, for which further research is recommended. First, this study focused on a convenience sample of two youths from the UO community with similar medical diagnosis and physical disabilities, using a single-subject design. Further studies are needed among a larger and more diverse population of participants with different types of disability, ages, and socio-cultural contexts, in order to enable better generalization of the findings. Second, the qualitative information was based on parents’ and youths’ reports; further studies can include separate reports. Finally, a long-term evaluation of this intervention is recommended, examining its longitudinal effects from the still-evolving COVID-19 situation. The effectiveness of the intervention can also be evaluated during the post-pandemic period and in different kinds of adverse situations.

## 5. Conclusions

This study provided initial evidence of the effectiveness of the PREP intervention approach for two youths from the ultra-Orthodox Jewish community during the COVID-19 pandemic. A research design using both quantitative and qualitative descriptive content analysis was used to enable integrated information, including a single-subject design with multiple baselines, and semi-structured interviews. The results indicated that, following intervention, the parents and youths reported improvements in their performance of all their selected goals and in their participation patterns, and high levels of satisfaction with the service received. As part of the intervention, barriers impacting participation and supporting factors (related to the participant, the activity demands, and the environment) were identified, along with environment-based strategies to support the intervention. The categories that emerged from the interviews provided additional information on the personal and environmental barriers, the factors supporting intervention, and the intervention effects. Additionally, the unique cultural codes and values of the UO community, as well as their low socioeconomic status, were considered during the intervention process.

Our results emphasize the need to consider the participants’ socio-cultural and physical environments in planning intervention, which may call for adaptation in order to promote participation in desired activities that truly matter to the participants within their natural environments. Since environmental factors can be changed, these also need to be addressed as part of intervention programs focused on promoting participation. The study results suggest that implementation of the intervention in light of changing needs and unexpected circumstances, such as the COVID-19 pandemic, require flexibility and creativity to jointly identify activities and seek solutions to make participation possible, including collaboration with other stakeholders as a team to support the intervention.

## Figures and Tables

**Figure 1 ijerph-20-03913-f001:**
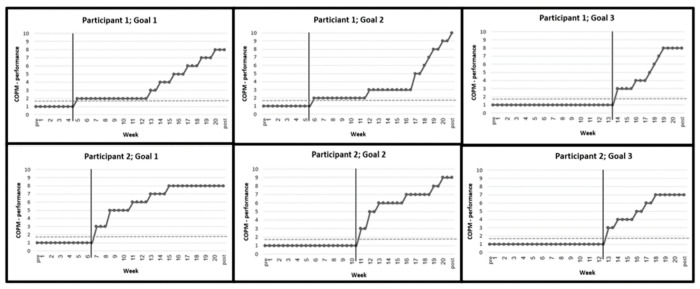
Changes in the Canadian Occupational Performance Measure (COPM): performance scores during baseline (week 1–4), intervention (week 5–16), and follow-up (17–20) phases. Notes: Participant 1: Goal 1: Buying sweets at the supermarket for the nephews; Goal 2: Preparing food for family members; Goal 3: Participation in prayers in the synagogue. Participant 2: Goal 1: Buying clothes with the sister; Goal 2: Preparing a dish for a family meal according to a recipe; Goal 3: Maintaining social interaction in meetings with peers.

**Table 1 ijerph-20-03913-t001:** Goals, examples of environmental barriers and intervention strategies for each participant.

Goals	Barrier		Environmental Intervention Strategy
Participant 1			
Goal 1: Buying sweets at the supermarket for the nephews	Activity: Activity demands	The participant lacked knowledge and experience in navigating within the supermarket, choosing the desired products, and managing money.	OT contacted city welfare department to recruit a tutor for practising shopping at the supermarket, deciding on criteria for selecting desired products, and practice in using money.
	Participant: Emotional ability	Participant has low motivation and is afraid of new experiences.	OT guided the tutor to support and encourage the participant in practicing this activity, in order to increase his motivation.
Goal 2: Preparing food for family members	Activity: Activity demands	Participant lacked prior knowledge how to organize preparation of food (cooking and baking).	OT guided the sister to rate the complexity of food preparation plans.
	Environment: Social support	Participant’s mother restrained him from engaging in food preparation activities.	OT recruited the participant’s older sister to allow exposure and encourage the participant to be involved in food preparation.
Goal 3: Participation in prayers in the synagogue	Participant: Emotional ability	The participant was afraid to leave his home and be among people.	OT contacted city welfare department to recruit a tutor for guidance and emotional support when leaving home for prayer at the synagogue.
	Environment: Physical environment	Parents did not know of a synagogue near the house that was accessible for a wheelchair.	OT encouraged parents to look for an accessible synagogue near their house.
		Participant was unfamiliar with safe crossing rules on the road.	OT guided the tutor to accompany the participant and practise with him how to safely cross the road.
Participant 2			
Goal 1: Buy clothes with the sister.	Activity: Activity demands	Participant lacked knowledge and had difficulty choosing suitable clothes and managing money.	OT coordinated between the participant and her sister, so that she can be guided during the purchase.
	Environment: Physical environment	Clothing stores near her home were not accessible for a wheelchair.	OT encouraged shopping in an accessible mall near her house.
Goal 2: Preparing a dish for a family meal according to a recipe.	Activity: Activity demands	Participant had difficulty preparing dishes according to recipes and needed detailed and orderly explanations.	OT shared information with family about use of strategies to organize the required information (via table) and writing recipes. She also taught the mother how to choose easy recipes with the participant.
	Environment: Physical environment	Products and kitchenware required to prepare food were not accessible to the participant seated in a wheelchair.	OT guided the family to organize and adjust the location of the required products and kitchenware in an accessible manner.
Goal 3: Maintaining social interaction in meetings with peers.	Participant: Cognitive ability	Participant had difficulty initiating social interaction in meetings or telephone conversations with peers.	OT guided participant in strategies to prepare for social interaction by thinking of potential conversation topics. She also contacted an art teacher to conduct a joint creative activity for the participant and other girls.
	Environment: Physical environment	Lack of accessible places for wheelchairs near the place of residence and at the friend’s house.	OT encouraged the participant and her family to initiate a weekly social gathering at her home.

**Table 2 ijerph-20-03913-t002:** Categories and sub-categories that emerged from the interviews.

Category	Sub-Category
Personal barriers	(a). Social barriers: Difficulties in communication with peers.
	(b). Emotional barriers: Feeling low self-efficacy and low motivation.
	(c). Cognitive barriers: Difficulties in understanding the activity’s demands.
Environmental barriers	(a). Physical barriers: Lack of accessibility to the physical environment in the community.
	(b). Social barriers: Attitudes of family and community members.
	(c). Cultural barriers: Customs of the UO community.
Factors supporting intervention	(a). Encouragement and support provided by the OT and others.
	(b). Mediating and imparting knowledge and strategies provided by the OT.
	(c). Accessibility and adaptation of the environment and/or the activity.
	(d). Practicing activities with others.
	(e). Acceptance by others.
Intervention effects	(a). Improved independence.
	(b). Improved initiative in activities.
	(c). Feeling happier and more enjoyment.
	(d). Improved confidence and a sense of competence.
	(e). Improved communication with others.

## Data Availability

Data is unavailable due to privacy or ethical restrictions.

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
