# Peer review of "Improving Participation among Youth with Disabilities within Their Unique Socio-Cultural Context during COVID-19 Pandemic: Initial Evaluation"

_ijerph, 2023, doi:10.3390/ijerph20053913_

Round 1

Reviewer 1 Report

Overall comment

·       Being sure to use children or youth with disabilities, and not individual with disabilities when the literature used focus on this population. For instance, see page 1, line 39 where use “individual” when the reference is about children and youth or page 2 line 49. The results and experience of participation are not the same.

Introduction

·       Please provide more details about PREP? Where was it developed? Where was it used? It’s describe as “One of the interventions that focuses on environmental factors », but we need more context. Does it focus on specific environment? The example of effectiveness presented seems to be for leisure or physical activities only.

·       The second paragraph mention high level of satisfaction with PREP. In what sense? How was that measured. Also the authors mention impact on specific functions without more specifics. Please be more detailed on this again. As this paper focus on the application of PREP (specifically in the context of COVID-19) and the claim made by the authors that “These 77 findings demonstrate the potential impact of the PREP approach to improve participation 78 among a population with diverse ages, disabilities, and living environments ».  , those details are really important.

·       The authors highlight the importance of looking at other socioeconomic background, but we don’t have the details of previous studies mention, what were the population? The explanation of why using the chosen population is great.

Methods

·       Please explain why the inclusion in a special education program was an criteria of inclusion for the participants

·       A reference for the CSW-8 validy and reliability is needed.

·       For the interview, the authors mention  a “participation team” who does it refer to? This information is given later in the process. Maybe consider changing order of presentation of some information.

·       Procedure section: I would suggest to move the recruitment procedure under participant sub section. It is currently confusing to read the information about the participants in another section. From the same procedure section, I would propose to present the procedure of the PREP before the questionnaire.

·       I would recommend moving the paragraph starting with “The SPSS version 26.0 (Statistical Package for the Social Sciences) [44] was used for the quantitative analysi »  under quantitative analysis subsection.

·       In the qualitative analysis, the authors mention using a thematic analysis but then explain mostly the creation of categories and sub-categories. How were the themes developed then? The results are more categories than thematic, consider revising the type of qualitative analysis chosen. This is not following proper Brown and Clarke thematic analysis methodology.

Results

·       Are the changes in the CEM-CY are clinically or statistically significant? The authors report a difference, but it is hard to assess what this difference really mean. Please specify.

·       The qualitative quotes are reporting narratives from different people such as sister or tutor. I figured that those are “participation team members”. Who are those people and how they were involved really need to be specified and explained better. Moreover a table of who of those were interviewed and maybe their charactheristics should be included in the paper as most of the qualitative results are coming from them.

·       Under  Factors supporting intervention » some quote talk about the therapist, is it the OT who did the PREP? Please specify.

·       The Table 2 formatting is really confusing. It is hard to see which sub-categories goes with which “theme”.

Discusison

·       On page 12, line 502-510, it is not clear how this entire paragraph is related to the results of the study, does it goes with the fact that the activity of leisure chosen were inside the home. Please consider merging the two paragraph.

·       Consider reviewing the last paragraph of the discussion, it is not clear what the author is trying to say.

Limitation

·       The second limitation mentioned was : “Second, the qualitative information was  based on parents' and youths' reports; further studies can include separate reports, as well as perspectives from other family and community members » but the participant included were tutor or sibling as well. So this limitation should be removed.

Conclusion

·       The conclusion mention well how this study despite being focused on a really specific population still provide clinical insights about both the PREP and clinical intervention in time of COVID

·        

Small details

Page 2 line 50-52. The sentence should read: Environmental support, however, was found to be significantly lower among children and youth with disabilities such as those with attention deficit hyperactivity disorder (ADHD), which especially impacted their involvement in different settings [13]

Reviewer 2 Report

This is a very well written manuscript which provides useful ideas on ensuring the successful implementation of PREP in a group likely to experience more intersectional challenges due to specific cultural considerations – during a time of more general societal restrictions (i.e. Covid).

The rationale for the study is strong. Please check throughout the introduction when evidence to support statements can be included. For example, in this sentence “A recent study [27] found that the culture also impacts the life experience and health outcomes of UO mothers raising children with ADHD.” I want to know how, in what ways. Please check through this section to ensure clarity with these sorts of statements throughout.

Please give some more context to this statement. The benefits of participation generally become more evident  during extraordinary times; yet the COVID-19 pandemic restrictions, which required staying more at home and following social distancing rules, added another layer of 105 complexity for children and youth with disabilities [28].  I think this is a too simplified viewpoint.

Methods:

What was the position of the research team to the research question and why mixed methods was appropriate.

Please discuss the type of mixed-methods in some more depth

Please clarify the extent to which the goals were participant or family driven. I am asking about the balance of contribution between the therapist, family, and participants in generating goals, identifying barriers and problem solving. Some discussion on your reflection on involvement/engagement with the process by the various parties in the discussion would be useful. It might be related to the cultural-social context or these individuals specifically.

What questions were asked by the interviewer. Were they structured, semi-structured. How was the interview schedule developed.

How relevant is CI to the data for interpreting the outcomes from N=2? Please clarify.

More information about content analysis is required.

The themes appear deductive, please explain some more how they were derived. This is another important reason for including the interview schedule.

As this is a mixed methods study, I would like some more information on how (and when) the data from the two aspects of the study were synthesised.

Discussion

What impact do you think Covid had on the outcomes in the end? The implementation of the intervention is discussed but not really the outcomes.

How feasible is this intervention more broadly. Thinking about cost and time and resources. This might be useful to others considering this concept.

Very well constructed conclusion.

Round 2

Reviewer 1 Report

Thank you for submitting this reviewed version. It answered all the comments I had. Please verify the font for the entire manuscript, many of the changes made are in a different size font.

Author Response

Reply to reviewer 1

The following is our detailed reply to the reviewers.

Comment #1: Thank you for submitting this reviewed version. It answered all the comments I had.

Response #1: Thank you for this comment.

Comment #2: Please verify the font for the entire manuscript, many of the changes made are in a different size font.

Response #2: Our revised manuscript included the correct font (it was submitted in Word version, as requested). However, it appears that the journal's editors altered the font in order to adapt this to a different format. The managing editor was updated, since changes could only be made by the editorial office.  

Reviewer 2 Report

Thank you for the amendments made. I think you could have kept the terminology of mixed-methods if you had wised as it feels like a mixed-methods study i.e, QUAN + QUAL but this just needed to be more firmly/clearly stated.

However, you have taken another approach, which is also valid. I suppose i was hoping the team would carefully consider their ontological and epistemological position to the research question, design and therefore interpretation. The new additions to the content analysis are helpful for readers.

All other queries have been addressed. Thank you.

Author Response

Reply to reviewer 2

The following is our detailed reply to the reviewers. In the revised manuscript we highlighted the changes in yellow.

Comment #1:

Thank you for the amendments made. I think you could have kept the terminology of mixed-methods if you had wised as it feels like a mixed-methods study i.e, QUAN + QUAL but this just needed to be more firmly/clearly stated. However, you have taken another approach, which is also valid. I suppose i was hoping the team would carefully consider their ontological and epistemological position to the research question, design and therefore interpretation. The new additions to the content analysis are helpful for readers.

All other queries have been addressed. Thank you.

Response #1:

Thank you for accepting the amendments made.

Following the first review we carefully discussed the terminology of mixed method, including consulting with another qualitative researcher and consulting the literature. As a result, we added an explanation which seems more accurate, indicating that descriptive content analysis was done. We also agree that a combination of quantitative and qualitative is valid and more accurately represent the study design and methodology used.

The research team considered the position to the research question, which was to evaluate the effectiveness of the PREP intervention. For this purpose, the study design included quantitative descriptive data through interviews with the participants and their participation team. Those interviews enable to receive additional information on their overall experience and perspectives specifically relating to the intervention process (e.g., different factors that support and hinder participation in selected activities and the intervention effect). This was further revised in the study design section (2.1).

We hope that this clarifies the approach we have taken. Please let us know if there is anything else we should add in the manuscript to further clarify this.